# Factory-Calibrated Continuous Glucose Monitoring Systems in Type 1 Diabetes: Accuracy during In-Clinic Exercise and Home Use

**DOI:** 10.3390/s23229256

**Published:** 2023-11-18

**Authors:** Sissel Banner Lundemose, Christian Laugesen, Ajenthen Gayathri Ranjan, Kirsten Nørgaard

**Affiliations:** Steno Diabetes Center Copenhagen, Clinical Research, Diabetes Technology Research, Borgmester Ib Juuls Vej 83, DK-2730 Herlev, Denmark; christian.laugesen@regionh.dk (C.L.); ajenthen.ranjan@regionh.dk (A.G.R.); kirsten.noergaard@regionh.dk (K.N.)

**Keywords:** type 1 diabetes, continuous glucose monitoring, exercise, accuracy

## Abstract

Continuous glucose monitors (CGMs) are valuable tools for improving glycemic control, yet their accuracy might be influenced by physical activity. This study sought to assess the accuracy of the three latest factory-calibrated CGM systems available in Europe at the time the study was conducted, both during aerobic exercise and in typical daily scenarios. The accuracy evaluation, based on metrics such as the median absolute relative difference (MARD) and point and rate error-grid analyses (PEGA and REGA), involved 13 adults with type 1 diabetes. Participants wore all sensors during a 1 h in-clinic exercise session followed by a subsequent 3-day home period, with blood glucose measurements serving as reference values in both contexts. During exercise, no statistically significant differences in MARD were observed (Dexcom G6: 12.6%, Guardian 4: 10.7%, and Freestyle Libre 2: 17.2%; *p* = 0.31), and similarly, no significant differences emerged in PEGA-zone-AB (100%, 100%, 96.8%; *p* = 0.37). Nevertheless, Freestyle Libre 2 showed comparatively diminished accuracy in estimating glucose trends during exercise (REGA-zone-AB: 100%, 93.0%, 73.3%; *p* = 0.0003). In the home environment, Freestyle Libre 2 exhibited a significantly higher MARD when compared to the other systems (10.2%, 11.9%, 16.7%, *p* = 0.02). Overall, Dexcom G6 and Guardian 4 demonstrated superior accuracy in both exercise and daily life scenarios compared to Freestyle Libre 2.

## 1. Introduction

Continuous glucose monitoring (CGM) supports diabetes management in individuals with type 1 diabetes (T1D) [1]. Most CGM systems measure glucose levels in the interstitial fluid using electrochemical subcutaneous sensors containing glucose oxidase. These sensors detect glucose levels in the interstitial fluid by catalyzing the oxidation of glucose, resulting in the generation of an electric current. This electric current is then transmitted to a receiver or monitor, providing a rich stream of glucose readings [2,3]. This wealth of data offers users and healthcare professionals a comprehensive perspective on glucose control, encompassing trends and patterns [3,4]. For the CGM to function optimally these devices must exhibit high accuracy, aiming to align the sensor’s data as closely as possible with the reference values, which are typically represented by plasma glucose (PG) or the self-monitoring of blood glucose (BGM). This crucial aspect of accuracy is often quantified using the Mean Absolute Relative Difference (MARD) [4,5], which serves as a measure for evaluating sensor performance. Notably, a MARD threshold of 10% has been suggested as a prerequisite for the direct use of CGM in making insulin dosage decisions [6]. However, accuracy during exercise is a challenge for CGM systems due to rapid changes in subcutaneous tissue circulation, blood oxygen concentrations, body temperature, sensor displacements, and glucose fluctuations [7,8,9,10].

A systematic review conducted by Moser and colleagues found that older versions of CGM systems had an average MARD of approximately 13.63% during exercise [4]. As technology continues to progress, the reduction of MARD is anticipated with the introduction of newer CGM generations [11]. The primary objective of this study was to conduct a comparative analysis of the accuracy demonstrated by the three latest factory-calibrated CGM systems available in Denmark at the time the study was conducted (Dexcom G6, Guardian 4, and Freestyle Libre 2) under two distinct conditions: a controlled in-clinic aerobic exercise session and a home monitoring period.

## 2. Materials and Methods

This study was part of a larger clinical study investigating the effect of low-dose glucagon to prevent exercise-induced hypoglycemia in people with type 1 diabetes. The study was conducted at the clinical research unit at Steno Diabetes Center Copenhagen, Herlev, Denmark. The study was monitored by the Good Clinical Practice Unit at Bispebjerg and Frederiksberg Hospital, Copenhagen, Denmark and approved by the Danish Medicines Agency, Copenhagen, Denmark (EudraCT 2021-001342-34, approved 12 July 2021), the Regional Committee on Health Research Ethics, Copenhagen, Denmark (H-21017717, approved 27 August 2021), and the Danish Data Protection Agency (P-2021-554, approved 17 September 2021). The study was registered at clinicaltrials.gov (NCT05076292) and conducted in accordance with the Declaration of Helsinki, Finland. 

### 2.1. Study Design

Participants were recruited from the outpatient diabetes clinic between September 2021 and February 2022; they underwent a screening visit, an in-clinic exercise session, and a subsequent 3-day monitoring period conducted at the participant’s home, including their free-living environment. Written informed content was obtained from all subjects involved in the study. During the screening visit, participants completed an International Physical Activity Questionnaire (IPAQ-SF) [12] to assess their physical activity levels. This information was used to determine whether they could engage in a 60 min bicycle exercise session.

The key inclusion criteria were age ≥18 years; duration of type 1 diabetes ≥2 years; use of an insulin pump or multiple injection therapy for ≥6 months; HbA1c ≤70 mmol/mol (8.5%); body mass index (BMI) ≤33 kg/m^2^; and engagement in exercise ≥1 h per week. The key exclusion criteria were the use of anti-diabetic medicine (other than insulin), high levels of activity (≥5 h of exercise per week) and known allergies to any of the CGM device patches. 

### 2.2. Insertion of the CGM

The three CGM devices (Dexcom G6 (Dexcom Inc., San Diego, CA, USA), Guardian 4 (Medtronic, Northridge, CA, USA), and Freestyle Libre 2 (Abbott, Chicago, IL, USA)) were subcutaneously inserted into the upper arm of each participant 48 h prior to the in-clinic exercise session. Freestyle Libre 2 and Guardian 4 were inserted on the same arm and Dexcom G6 on the other arm. In most cases, the participant already wore one of the three CGM devices and thereby decided on the placement of the other ones. The readings from the Dexcom G6 and Freestyle Libre 2 were transmitted to their respective receivers, whereas data from the Guardian 4 was routed to a disconnected MiniMed^TM^ 780G insulin pump. Throughout the entire trial, none of the CGM devices underwent calibration or were blinded. 

### 2.3. In-Clinic Session

The participants arrived at the research facility at ∼8 AM. They initiated a 60 min exercise session of moderate intensity (50% of each participant’s estimated heart rate reserve) [13] on an ergometer cycle (Lode Corival CPET, Groningen, The Netherlands). This exercise session was followed by a 2 h resting period before concluding the study visit. During the exercise session, PGM data were collected and analyzed using a YSI (YSI 2300 STAT+, Yellow Springs, OH, USA) at 5 min intervals. For the resting phase, PGM data were collected every 15 min due to the expectation of lower glucose fluctuation. BGM was performed every 10 min during the exercise session using a Contour Next One (Ascensia, Baal, Switzerland) [14] blood glucose meter. 

The collected PGM values were utilized in the larger clinical study. Results pertaining to the accuracy of the sensors, using PGM as a reference, are provided in the Appendix A and will not be addressed in the main article. 

### 2.4. Home Period

The participants continued wearing the three sensors for an additional three days during the home period. They were instructed to conduct BGM assessments at least five times daily—i.e., before main meals and once in the afternoon and evening, with intervals of at least 60 min between each measurement—using the Contour Next One meter. The BGM values were not used for treatment decisions and were not to be repeated in case of deviation from the CGM value. To gain an overview of the trial days, please see Figure 1.

### 2.5. Outcomes and Data Definitions

The primary outcome was the difference in MARD [15] between the three CGMs during exercise, using BGM as the reference value. Secondary outcomes included point error grid analysis (PEGA), rate error grid analysis (REGA), precision absolute relative deviation (PARD), and MARD during the 3-day home period.

To evaluate the MARD, each BGM measurement was paired with its closest-in-time CGM value (≤3 min apart) for each of the three sensors. Identical matching of BGM–CGM pairs was performed for the three days at home.

The rate of change was determined by finding the per-minute glucose concentration for each of the two methods, i.e., CGM and BGM. To compute the glucose concentration per minute, we divided the change in glucose concentration by the time elapsed between two consecutive measurements. Subsequently, the plotted comparison of BGM rates against CGM rates was classified into five zones of the REGA scatterplot (labeled A to E), where AB marked the zone considered to be accurate. The frequency at which the CGM-measured rate of change matched the BGM-measured rate of change was evaluated. The PEGA corresponds to the traditional Clarke-Error Grid, where zone AB was considered to be accurate [16].

PARD was assessed by parring CGM measurements (with a maximum interval of 3 min) across the three devices. The Dexcom G6 and the Guardian 4 provided CGM readings every 5 min, while the Freestyle Libre 2 offered readings with 15 min intervals.

### 2.6. Statistical Analysis

Calculation of MARD, PARD, PEGA zone AB, and REGA zone AB was performed for each CGM system for each participant. To test significant differences in MARD between the sensors, a repeated measurement ANOVA was performed, with participants treated as random effects and the sensors (Dexcom G6, Guardian 4, and Freestyle Libre 2) as the fixed effect. If there was an overall device effect, a pairwise comparison between sensor devices was performed, adjusting for multiple comparisons using the Bonferroni method. Before conducting the error grid analyses, we utilized a Bland–Altman plot to evaluate the bias of the sensors and employed a scatter plot to evaluate Pearson correlations. A Cochran’s Q test for heterogeneity was used to compare the PEGA and REGA proportion between the sensors. If Cochran’s Q test showed an overall significant difference in the proportions of zone AB, a post-hoc analysis with a McNemar test was performed with a Bonferroni correction. Statistical analyses were performed with SAS 9.4 (SAS Institute, Inc., Cary, NC, USA). *p* values < 0.05 were considered statistically significant.

## 3. Results

### 3.1. Baseline Characteristics

A total of thirteen participants went through screening (10 men and 3 women). One participant was excluded from the analysis due to missing sensor values from two of the CGMs, leaving a cohort of 12 participants for the subsequent analysis. The baseline characteristics of the participants are as follows: mean (min–max) age 58 years (46–78), BMI 26 kg/m^2^ (21–30), duration of T1D 37 years (8–63), HbA1c 56 mmol/mol (42–66)/7.3% (6.0–8.2), and total daily insulin dose 43 IU (14–70).

### 3.2. Mean Absolute Relative Difference

There were a total of 79 BGM measurements during exercise and 215 BGM measurements in the home period that could be used as a reference for the three CGM devices. For an overview of data collected for the CGMs in the different periods, please see Table 1.

No significant difference in MARD was found between the three CGM systems during the exercise session when using BGM as the reference measurement (Dexcom G6: 12.6%, Guardian 4: 10.7% and Freestyle Libre 2: 17.2%; *p* = 0.31). However, during the home period, the MARD for Freestyle Libre 2 (16.3%) was significantly higher than for Dexcom G6 (10.2%) or Guardian 4 (11.9%); p_overall_ = 0.02 (Table 2).

### 3.3. Error Grid Analysis

A Bland–Altman plot revealed a minor positive bias for Dexcom compared to BGM during exercise, while Guardian showed a negative bias. Both sensors exhibited a limit of agreement (LoA) within +/−2 mmol/L, whereas Libre displayed a wider variability, with an LoA spanning from 3 to 4 mmol/L (Appendix A). The results from the PEGA are presented in Table 2 and Figure 2 and Figure 3. During the exercise session, there were no significant differences between the combined readings in zone A and B for any of the CGMs (Dexcom G6: 100%, Guardian 4: 100%, and Freestyle Libre 2: 96.8%; *p* = 0.37), using BGM as the reference. For the home period, the PEGA Zone AB was comparable between the sensors (Dexcom G6: 97.8%, Guardian 4: 98.0%, and Freestyle Libre 2: 99.0%; *p* = 0.42).

Figure 2 Point Error Grid for the exercise setting.

**Figure 2 sensors-23-09256-f002:**
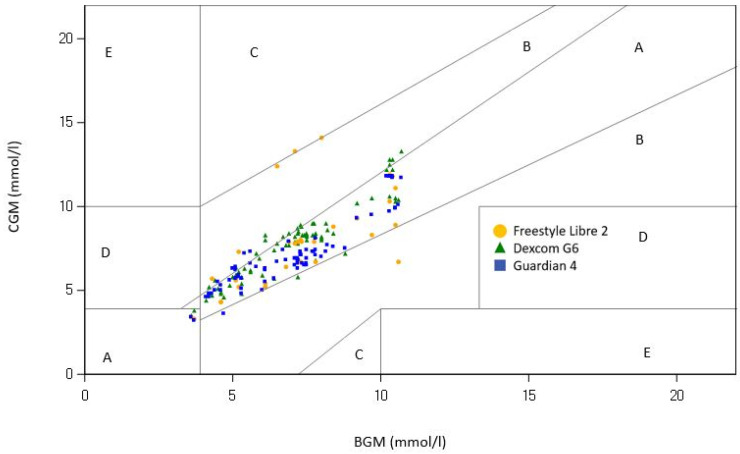
Point Error Grid Analysis for the exercise setting using BGM (blood glucose measurement) as a reference. Three continuous glucose monitoring systems were used (Freestyle Libre 2 = yellow, Dexcom G6 = Green, Guardian 4 = Blue). The grid is divided into zones signifying the degree of risk posed by the incorrect measurement: zone A represents no effect on clinical action; zone B represents altered clinical action—little or no effect on clinical outcome; zone C represents altered clinical action—likely to affect clinical outcome; zone D represents altered clinical action—could have significant medical risk; and zone E represents altered clinical action—could have dangerous consequences.

Figure 3: Point Error Grid for the home period.

**Figure 3 sensors-23-09256-f003:**
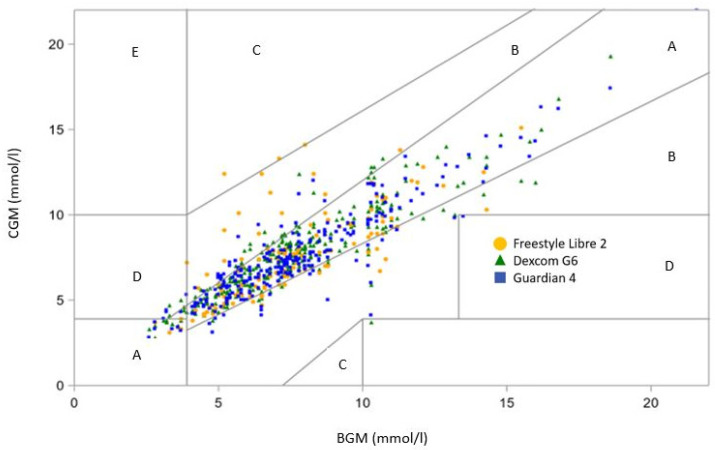
Point Error Grid Analysis for the home period using BGM (blood glucose measurement) as a reference. Three continuous glucose monitoring systems were used (Freestyle Libre 2 = yellow, Dexcom G6 = Green, Guardian 4 = Blue). The grid is divided into zones signifying the degree of risk posed by the incorrect measurement: zone A represents no effect on clinical action; zone B represents altered clinical action—little or no effect on clinical outcome; zone C represents altered clinical action—likely to affect clinical outcome; zone D represents altered clinical action—could have significant medical risk; and zone E represents altered clinical action—could have dangerous consequences.

During exercise, the mean Rate Of glucose Changes (ROC) was 1.25% ± 1.25. In REGA zone AB, using BGM as a reference, the ROC was 100% for Dexcom G6, 93% for Guardian 4, and 73.3% for Freestyle Libre 2; p_overall_ = 0.0003 (p_(Dexcom G6- Freestyle Libre 2)_ = 0.023, p_(Dexcom G6-Guardian 4)_ = 0.09, and p_(Freestyle Libre 2-Guardian 4)_ = 0.023).

### 3.4. Precision Absolute Relative Difference

The PARD between the three sensors during exercise was 13.6% for Dexcom G6–Freestyle Libre 2, 11.4% for Dexcom G6–Guardian 4, and 13.1% for Guardian 4–Freestyle Libre 2. PARD in the home setting was 16.4% for Dexcom G6–Freestyle Libre 2, 10.4% for Dexcom G6–Guardian 4, and 14.2% for Guardian 4–Freestyle Libre 2.

## 4. Discussion

The aim of this study was to assess the accuracy of three calibration-free CGM/isCGM systems (Dexcom G6, Guardian 4, and Freestyle Libre 2) in individuals with type 1 diabetes across both an aerobic exercise session and a subsequent home period. The outcomes demonstrated no significant difference in MARD or point accuracy between the three sensors during exercise, when using BGM as the reference measurements. Similar results for the exercise session were also observed when employing PG as the reference; details can be found in the Appendix A, Appendix A. Nonetheless, the trend accuracy during exercise was significantly lower for Freestyle Libre 2 compared to Dexcom G6 and Guardian 4. Additionally, during the at-home period, Freestyle Libre 2 showed a significantly higher MARD compared to the other sensors.

While the accuracy of different CGMs has been widely studied [1,16,17,18,19,20], to our knowledge, this is the first study to compare the accuracy of Guardian 4 against other calibration-free CGM/isCGM systems (Dexcom G6 and Freestyle Libre 2) under varying conditions. The main strength of this study lies in the concurrent wearing of all three sensors within the same region, with each being compared to an identical reference measurement.

The findings for Dexcom G6 and Freestyle Libre 2 remain consistent with previous studies. For instance, Guillot et al. [17] reported that Dexcom G6 accuracy remained unimpaired during aerobic exercise, with a MARD range of 8.9 to 13.9%. Similarly, Aberer et al. [18] found a MARD of 8.7 (±5.9)% for the Freestyle Libre during exercise. However, Da Prato et al. [1] reported a different MARD of 44.9 (±14.4)% for the Dexcom G6 during continuous exercise compared to interval training, while Moser et al. [21] identified a MARD value of 30 (18–40)% for the Freestyle Libre during exercise compared to 14 (7–23)% during home use.

Interestingly, our study did not identify any difference in MARD between exercise and the home period, but the variation of MARD was numerically higher during exercise compared with the home period.

This study is limited by its modest sample size. The considerable standard deviation of MARD values during the exercise session might be reduced by including more participants, and thus potentially yield statistically significant outcomes. A power analysis to determine the sample size or to pre-determine the primary outcome measures would have been preferable. Moreover, the study design has inherent constraints, including the limited collection of BGM data points and the presence of missing BGM–CGM reference pairs both during exercise and in the home setting, restricting the calculation of MARD across different levels. An improvement could be to extend the monitoring phase at home and increase the number of BGMs to get a more precise estimate of MARD for the CGMs compared. Another limitation arises from the type and frequency of exercise, which was confined to be a single occurrence of moderate-intensity exercise. It is also important to acknowledge that the CGM devices were not used in a blinded manner, which could potentially impact the study results. However, blinding the Guardian 4 device was not feasible. As previously mentioned, this challenge was addressed by instructing participants not to duplicate their blood glucose measurements if they noticed variations from the values indicated by the CGMs.

Lastly, the development of CGM systems is progressing faster than studies are being conducted; there may already be newer and perhaps more accurate versions of both Libre and Dexcom (Freestyle Libre 3 and Dexcom G7) at this time.

## 5. Conclusions

This study indicates that, in comparison to Dexcom G6 and Guardian 4, Freestyle Libre 2 was less accurate in measuring glucose trends during exercise and less accurate in measuring glucose levels during daily activities.

## Figures and Tables

**Figure 1 sensors-23-09256-f001:**
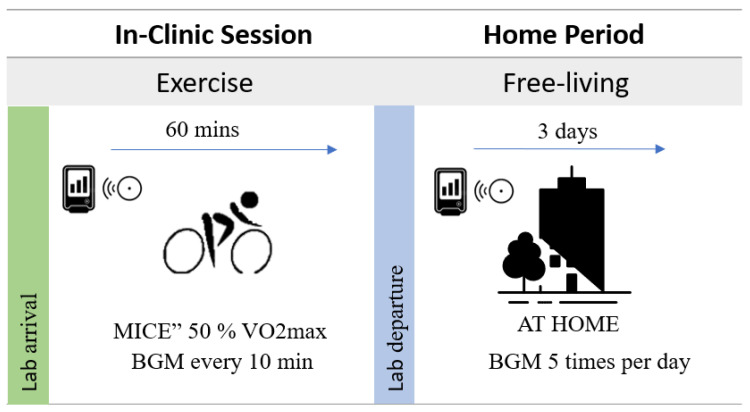
Overview of the trial days, i.e., In-clinic exercise session and home period.

**Table 1 sensors-23-09256-t001:** Number of participants, CGM measurements per participant, CGM–reference pairs and missing pairs in total for the three devices in the exercise and home period.

	Exercise				Home			
CGM	N Participants	CGM Measurements Per Participant (Mean)	CGM-Reference Pairs	N Missing	N Participants	CGM Measurements Per Participant (Mean)	CGM-Reference Pairs	N Missing
Dexcom G6	13	13	79.0	0.0	13	932	215.0	−1.0
Guardian 4	12	12	73.0	−6.0	12	850.2	192.0	−24.0
Freestyle Libre 2	12	4.8	31.0	−48.0	12	318.9	103.0	−113.0

**Table 2 sensors-23-09256-t002:** MARD, REGA, and PEGA results for the in-clinic exercise session and home period.

Settings	Variable	CGM			*p*-Value			
		Dexcom G6	Guardian 4	Freestyle Libre 2	Overall	Dexcom G6-Freestyle Libre 2	Dexcom G6- Guardian 4	Freestyle Libre2- Guardian 4
In-clinic exercise session	MARD ^a^ % ± SD	12.6 ± 5.7	10.7 ± 6.0	17.2 ± 15.6	0.31			
	REGA ^b^ AB%	100.0	93.0	73.3	0.0003 ^CQ^	0.023	0.09	0.023
	PEGA ^c^ AB%	100.0	100.0	96.8	0.37 ^CQ^			
Home period	MARD ^a^ % ± SD	10.2 ± 2.5	11.9 ± 4.1	16.3 ±9.3	0.022	0.0075	0.42	0.048
	PEGA ^c^ AB%	97.8	98.0	99.0	0.42 ^CQ^			

The standard deviation (SD) was computed from the MARD across participants. CQ: Cochran’s Q test for heterogeneity in proportion of REGA and PEGA AB between the three sensors. ^a^ MARD (Mean Absolute Relative Differences), ^b^ REGA (Rate Error Grid Analysis), and ^c^ PEGA (Point Error Grid Analysis).

## Data Availability

Data is available upon request due to restrictions for privacy and ethical reasons.

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
