# Peer review of "Factory-Calibrated Continuous Glucose Monitoring Systems in Type 1 Diabetes: Accuracy during In-Clinic Exercise and Home Use"

_sensors, 2023, doi:10.3390/s23229256_

Round 1

Reviewer 1 Report

Comments and Suggestions for Authors

The manuscript investigates the accuracy of three factory-calibrated CGM devices (i.e., Dexcom G6, Abbott Freestyle Libre 2, and Medtronic Guardian 4) in comparison to blood glucose concentration values collected with a fingerpick, during an exercise session performed in-clinic and 3 days of at-home monitoring.

While the topic is interesting, the manuscript lacks major details across all sections.

-       The introduction is very short, and it lacks important details about the relevance of this work.
Why is CGM important in individuals with type 1 diabetes? Why evaluating the performance of CGM during exercise is particularly relevant for people living with diabetes?
The authors cite a review paper about the accuracy of old generation CGM devices, but an overview of the tested (new generation) devices is completely missing: what is the state-of-the-art about their accuracy (overall and during exercise, if available)? Why did the author choose exactly these 3 devices?

-       In the methods, the study design lacks crucial information, affecting the interpretability of the results (see comments below). Some choices related to the analysis are not documented. In addition, a presentation of the assessment metrics and how they are summarized is missing. Finally, a more comprehensive assessment is recommended, including, for example, the study of Pearson’s correlation, Bland-Altman analysis, and interclass correlation coefficient between glucose measurements collected from test CGM devices and reference comparator.

-       Results are reported in a chaotical way, sometimes ambiguous, and they are inconsistent throughout the text (i.e., different numbers reported in the text, Table 1, and figures). Results about the distribution of the paired test-reference measurements, as well as the compliance and data completeness during the at-home portion are missing.

-       The discussion might be revised based on the above and below comments, and some study limitations should be acknowledged. The reference list is extremely poor for a topic which has been widely studied in the literature. See some suggestions below.

Major and minor comments/suggestions are detailed below.

Based on the above and below comments, this article requires extensive revisions in terms of quality of presentation, methodological rigor, and presentation of the results; therefore, a publication in its present form is not recommended.

Major comments:

-       Introduction, lines 44-45. “The latest factory-calibrated CGM systems may have a lower MARD during exercise”. Where does this claim come from? The authors might want to add some references about the evaluation of the accuracy of CGM devices during exercise for new generation CGM devices (see, for example, https://doi.org/10.3390/s21020479, https://doi.org/10.1016/j.diabres.2015.11.012, https://doi.org/10.3390/bios8010022) or references about how the accuracy of new generation CGM devices generally improves as compared to old generation ones (see, for example, https://doi.org/10.1089/dia.2013.0077, https://doi.org/10.1089/dia.2009.0075)

-       Study Design: Was the intensity of the physical activity task self-chosen by participants? Did they receive specific instructions on how to keep the same pace?

-       Study Design: The physical activity session lasts 60-min, what about those participants not being able to complete all 60 min?

-       Study Design: Were participants asked to stop the task to collect PGM and BGM measurements?

-       Study Design: Are the YSI 2300 STAT+ and the Contour Next One cleared by regulatory agencies to be used as reference devices?

-       Study Design: the design of the overall study is reported, which includes 3 days of at-home monitoring. How is this choice justified? The authors might want to cite here, or in the limitations, that 15-17 monitoring days are recommended for a precise estimation of time-in-range metrics from CGM devices in type 1 diabetes (see, for example, https://doi.org/10.1111/dom.14483)

-       Materials and Methods, lines 94-54: Why is the sampling period from the reference device lower during resting? Is this related to an expected lower variability in the glucose fluctuations? The authors should clarify this choice.

-       Materials and Methods: Include model and brand of the ergometer cycle used in this study.

-       Materials and Methods, lines 104-105: Does “identical matching” mean that the pairing was performed only for those values with the exact same timestamp? Is that on a minute level? Why this choice? This might hardly reduce the number of paired samples, also given the different sampling periods across CGM devices. In addition, I would expect the tolerance on the time difference between CGM and BGM to be higher during at-home exercise compared to in-lab exercise, since glucose might vary faster during exercise.

-       Materials and Methods, line 105. The authors should clarify how the per-minute glucose concentration was derived from the different methods, with different sampling periods. Was it estimated as the ratio between difference in glucose readings and difference in time, or was an interpolation algorithm implemented?

-       Materials and Methods, lines 118-121. What covariates were used in the model? What type of ANOVA was used (type I, II, III?)? The authors should specify what the post-hoc analysis aims to assess.

-       Materials and Methods. “The study was part of a larger clinical study … [6]”. Reference 6 cannot be found online. The authors should include the NTC number, and explain whether the participants analyzed in this study represent only a subset of the overall dataset. If yes, how was this subset selected?

-       Materials and Methods. More details on the assessment metrics should be reported. For example, how is MARD computed? A formula could help clarifying. How is MARD summarized? First per participant and then across participants, or across all measurements?

-       Results, line 142. Two different values are reported for MARD for Dexcom G6 during the at-home period: 10.6% at line 142, 10.2% in Table 1.

-       Results, line 134. How are the BGM-CGM pairs distributed across devices? Are they balanced across devices? A description of the data gaps both during the in-lab session and for the at-home portion is completely missing. Similarly, how are the BGM-CGM pairs distributed across participants?

-       Results, line 135-136. “Missing BGM-CGM pairs for the Guardian 4 was for exercise period -48.0”. What are the authors computing here?

-       Results: The compliance in performing BGM at-home should be discussed.

-       Results, Table 1. N (number of paired measurements) is missing.

-       Results, Table 1. How is the standard deviation derived? Is it the standard deviation of the relative discrepancy, or the standard deviation of MARD across participants?

-       Results, line 152. Here, and in many other occasions throughout the manuscript, results are reported for the 3 test devices, but it is not specified which number belongs to which device, thus making hard to understand the conclusions.

-       Results, Figure 1 and 2. Figures annotations report number of participants equal to 13, but in line 129 the authors write that only data from 12 participants have been analyzed.

-       Discussion: Limitations of the use of unblinded CGM for at-home use should be added. Could participants be attempted to re-check their BGM if they saw a different value than that observed from CGM devices? Were they instructed to avoid this?

Minor comments:

-       Abstract, line 15. The authors refer to precision here, but accuracy would be more appropriate.

-       Abstract: why is the brand of the device only reported for the G6 (Dexcom), and not for the other devices?

-       Abstract, line 28. The authors should use Freestyle Libre 2 consistently throughout the manuscript.

-       Materials and Methods, line 64. Were participants asked to wear the device only while at home? If not, I recommend using a different wording, such as participants’ free-living environment, or at least to clarify that participants’ home includes their free-living environment.

-       Materials and Methods, line 75. Dominant or non-dominant arm? The authors should also define if the height of the CGM devices on the arm was consistent across individuals.

-       Materials and Methods, line 82. “physical activity” session

-       Materials and Methods, line 82. There is a bracket opened by not closed. The authors might want to discuss more how reference 7 contributed to the design of the physical activity task.

-       Results, line 130. Does “range” refer to the interquartile range or the minimum-maximum range?

-       Results, line 202. ROC should be defined at its first instance in the text.

-       Discussion, line 218. Substitute “result” with “results”.

Comments on the Quality of English Language

Overall, the quality of English language is acceptable. Some typos have been included among the comments.

Reviewer 2 Report

Comments and Suggestions for Authors

The submitted manuscript is a well-written, informative, and concise clinical study that compares several metrics of calibration free CGMs: Libre 2, Guardian 4, and G6. Of interest is that the Libre 2 appeared to underperform during the exercise and home measurement periods. This may be attributed to Libre's 2 electrode design. Also of interest is the inclusion of the Guardian 4, which the author mentions is the first study to do so in comparison to the other 2 CGMs. The findings suggest that the G6 and Guardian 4 may present more accurate data than the Libre 2, however without a larger sample size this is not absolutely statistical.

Limitations of the study include a small sample size, limited collection of BGM reference data points, and missing reference pairs. However, as clinical studies in the realm of diabetes care go, this is common  due to the age of participants. 

Reviewer 3 Report

Comments and Suggestions for Authors

In this work, authors proposed a study on Factory-Calibrated Continuous Glucose Monitoring Systems in 2 Type 1 diabetes: Accuracy During In-Clinic Exercise and Home 3 Use. 

Though the study (as claimed) is first of its kind but it needs to be improved. 

1. The experimental figures are not included in the paper which should be there to validate the study.

2. The authors used sensors as per the paper. It is necessary to describe and show these sensors in the paper.

3. More references are required to get a clear background information of the topic to the readers.

4. On page-4, Table-1 caption is separated from the table.

5. The font and font size of the text used in the figures 1 and  2 can be increased for better visualization and should be consistent.

Round 2

Reviewer 1 Report

Comments and Suggestions for Authors

The authors addressed most of the comments, and the quality of the manuscript has clearly improved.

Nonetheless, there are some concerns related to the newly added analyses:

- Table 1: despite the 79 available BGM measurements, for the FreeStyle Libre 2, a total of 12x4.8=57.6 CGM measurements are available. Is the number of CGM measurements lower by design (i.e., expected) - if yes, why? Or is there an unexpected loss of CGM data? - if yes, please argument.

- Despite Bland-Altman plots and scatter plots have been added to the Supplemental Material, they are barely mentioned in the text. However, for the sake of interpretability, they should be disclosed in the methods, and the resulting considerations should be reported in the manuscript.

- How were participants randomized to different device-arm groups? I.e., how the arm for each device was decided per each participant? The dominant/non-dominant arm might notably affect the quality of data (e.g., compression artifacts, movement, sweating, etc.), and consequently the measurements accuracy. If no formal randomization was performed, please explain how the assignment was decided, and add the lack of randomization in the study limitations. 

- "Participants completed a questionnaire to assess their physical activity level", please add details about the questionnaire (how many questions? Was it validated or used in previous studies?), and how the responses were used for the design.

- I strongly recommend careful review of the manuscript to avoid typos, such as "paticipant" (Table 1), "was inserted" (line 91).

Comments on the Quality of English Language

English quality is good.

Reviewer 3 Report

Comments and Suggestions for Authors

Authors have answered all my questions. It can be accepted now.

Author Response

Dear Reviewer,

I appreciate your feedback and the acceptance of the manuscript.

Round 3

Reviewer 1 Report

Comments and Suggestions for Authors

The authors addressed all comments/suggestions, and the paper can be published in its current form.

Comments on the Quality of English Language

No comments.